# A Novel Physical Mechanism to Model Brownian Yet Non-Gaussian Diffusion: Theory and Application

**DOI:** 10.3390/ma15175808

**Published:** 2022-08-23

**Authors:** Francisco E. Alban-Chacón, Erick A. Lamilla-Rubio, Manuel S. Alvarez-Alvarado

**Affiliations:** 1Faculty of Natural Science and Mathematics, Escuela Superior Politécnica del Litoral, Guayaquil 090112, Ecuador; 2Facultad de Ciencias Matemáticas y Físicas, Universidad de Guayaquil, Guayaquil 090514, Ecuador; 3Faculty of Electrical and Computer Engineering, Escuela Superior Politécnica del Litoral, Guayaquil 090112, Ecuador

**Keywords:** Lennard-Jones potential, phase separation, Brownian motion, non-Gaussian, molecular interactions

## Abstract

In the last years, a few experiments in the fields of biological and soft matter physics in colloidal suspensions have reported “normal diffusion” with a Laplacian probability distribution in the particle’s displacements (i.e., Brownian yet non-Gaussian diffusion). To model this behavior, different stochastic and microscopic models have been proposed, with the former introducing new random elements that incorporate our lack of information about the media and the latter describing a limited number of interesting physical scenarios. This incentivizes the search of a more thorough understanding of how the media interacts with itself and with the particle being diffused in Brownian yet non-Gaussian diffusion. For this reason, a comprehensive mathematical model to explain Brownian yet non-Gaussian diffusion that includes weak molecular interactions is proposed in this paper. Based on the theory of interfaces by De Gennes and Langevin dynamics, it is shown that long-range interactions in a weakly interacting fluid at shorter time scales leads to a Laplacian probability distribution in the radial particle’s displacements. Further, it is shown that a phase separation can explain a high diffusivity and causes this Laplacian distribution to evolve towards a Gaussian via a transition probability in the interval of time as it was observed in experiments. To verify these model predictions, the experimental data of the Brownian motion of colloidal beads on phospholipid bilayer by Wang et al. are used and compared with the results of the theory. This comparison suggests that the proposed model is able to explain qualitatively and quantitatively the Brownian yet non-Gaussian diffusion.

## 1. Introduction

The dynamics of Brownian diffusion are frequently used for modeling stochastic motions to obtain information about the particle’s interaction with binding partners and the local environment [1,2]. The main characteristic of Brownian diffusion lies in the feature of random wiggling particle motion that generally produces a normal or Gaussian distribution in the particle density function, with mean μ=0 and variance $\sigma^{2}=2Dt$, where *D* is interpreted as mass diffusivity or diffusion coefficient [3,4]. The Brownian diffusion model is very useful to analyze and study a variety of physical processes related to mechanisms of particle transport [5,6], thermal fluctuations [7,8,9], particle manipulation [10,11,12], and stellar dynamics [13,14]. Despite this Gaussian behavior being common in the displacement distribution for systems that exhibit Brownian motion, in the last years, efforts have been made to study a new type of Brownian diffusion. Like normal diffusion, it has a linear time dependence of the mean-square displacement (MSD), but is accompanied by a non-Gaussian displacement distribution, which has been identified as “anomalous yet Brownian” diffusion [15]. The Brownian yet non-Gaussian diffusion reported by Wang et al. is based on the classical random walk, in which mean-square displacement is simply proportional to time, but instead, the displacement distribution has an exponential behavior with the decay length of the exponential being proportional to the square root of time. This concept was vastly used to propose a model based on anomalous, but Brownian diffusion to describe the nature of diffusivity memory, but not the memory in the direction of the particle’s trajectories. The model was coined as diffusing diffusivity, due to the random walk that the diffusivity experiences [16]. Chubinsky–Slater’s idea in [16] has also been useful to model the behavior of biological, soft, and active matter systems establishing a minimal model framework of diffusion processes with fluctuating diffusivity [17]. Further, in the realm of fluids, it has been shown that in some confinement conditions, density fluctuations might be relevant to Brownian yet non-Gaussian diffusion [18,19].

The literature presents different kind of stochastic models that explain mathematically the Brownian yet non-Gaussian diffusion process. This includes studies of the role of media heterogeneity by randomizing parameters that appear in diffusivity dynamical equations [20]; demonstrations on time-dependent diffusivity, induced by external nonthermal noise [21]; and interesting comparisons between non-Gaussian random diffusivity models [22].

Microscopic models have also been proposed, where they have rigorously, mathematically studied interesting physical scenarios such as the study of diffusion of ellipsoidal particles, active particles, diffusion of colloidal particles in fluctuating corrugated channels, and Brownian motion in arrays of planar convective rolls [23]; non-Gaussian diffusion in static disordered media via a quenched trap model, where the diffusivity is spatially correlated [24]; and the Hitchhiker model [25].

All the exposed models have in common that they reflect different levels of our lack of information/ignorance about the surroundings/media. The lack of information about the media is considered by introducing random elements to the models via random parameters, random diffusivity, noise, or by introducing additional physical assumptions (i.e., either about the environment or the Brownian particle (i.e., large particle being diffused in the fluid) that lead to study interesting, but more complex physical scenarios.

From a practical point of view, there is plenty of evidence that these models give correct predictions. Nevertheless, from a theoretical perspective, to the best of our knowledge, a simple physical mechanism that incorporates media information by establishing a link between weak molecular interactions, phase separation, and Brownian motion has not been studied. This fact motivates the study and verification of model calculations that consider long-range Brownian particle–fluid molecular interactions, which via a phase separation (in a compressible fluid) attempts to explain the Brownian yet non-Gaussian diffusion by providing accurate predictions (including the experimental observation of the transition to a Gaussian process as observed in [15]).

The rest of the paper is structured as follows: Section 2 presents the theoretical framework that explains how two phase-separated clusters (i.e., a group of phase-separated fluid molecules and Brownian particle is denoted by “cluster”) interact. Section 3 discusses relevant experimental information and how it compares to the proposed model. Section 4 provides experimental data and quantitatively evaluates the predictions of the model. Finally, Section 5 incorporates the conclusion.

## 2. Mathematical Model of Two Interacting Phase-Separated Clusters

To explain the results of recent experiments [15,26,27,28], a model that considers molecular interactions during Brownian motion is proposed. Motivated by [10], it is hypothesized here that due to interactions between molecules of the Brownian particle and active components of the biological fluid, the Brownian particle becomes weakly polarized. Then, it can be said that if the magnitude of the attractive energy produced by molecules of the Brownian particle is high enough to break weak interactions between components of the biological fluid, a phase separation occurs [29]. Note that if these interactions are too weak, no phase separation happens and the fluid is dispersed (see Figure 1). In this case, regular Brownian motion as described by Einstein takes place.

The case of interest in this manuscript is the one with a phase separation; therefore, a model is proposed for this case. This is characterized by a highly compressible liquid phase (i.e., weakly interacting liquid, behaving as a gaseouslike state) that is, on average, separated (however, mixed) of a nearly incompressible liquid phase. (i.e., gel-like characteristics) [30]. Further, since in the gaseouslike state particles can move more freely than in the gel-like state, equilibrium is reached faster [29]. To incorporate this fact in the model, a shorter time scale τ, in which the system in the gaseouslike state relaxes to a local equilibrium is introduced (where no fluctuations occurs and local average of observables satisfy the ergodic hypothesis), where eventually at equilibrium, the $\tau_{e}$ (this is the experimentally observed time scale, where $\tau_{e}=t$) scale is reached (there are two scales in time for this phase). For the gel-like state, it is assumed that in the $\tau_{e}$ scale, the system has not yet reached equilibrium; therefore, there is only one scale in this state.

It is mentioned here that for clarity purposes, we reproduce the derivation from De Gennes [31]. Before starting this derivation, we note that more information on the variables used in the derivation and in the rest of the paper can be found in Appendix A. Now, to model the weak, long-range, and pairwise interactions between molecules of the Brownian particle and fluid molecules (i.e., dipole–dipole/Van der Waals interactions) the attractive part (long-range) of a 6-12 Lennard-Jones potential $U\left(r_{ij}\right)$ [32] is introduced, which is
(1)$$U\left(r_{ij}\right)=-4\varepsilon {\left(\frac{\sigma }{r_{ij}}\right)}^{6}$$
where ε is the dispersion energy, σ is the minimum distance a fluid molecule can be located with respect to a molecule at the interface (i.e., Van der Waals radius), and $r_{ij}$ is the distance between a molecule located in the Brownian particle and a fluid molecule.

Moreover, the Brownian particle is considered as a spherical interface interacting with the fluid, attracting fluid molecules. This model was already studied for a spherical shell interface in [31] and it was found that for the planar limit at $\left|R-r_{m}^{\prime }\right|\ll R$, with *R* being the radius of the spherical shell and $r_{m}^{\prime }$ the distance from the center of the sphere, a 10-4 Lennard-Jones potential emerges; therefore, for a sphere, a 3-9 Lennard-Jones potential [31] would follow, where we are paying attention only to the cubic decaying part, as stated above. The limit we are interested in is the opposite limit, which is the long-range limit of $\left|R-r_{m}^{\prime }\right|\gg R$. In this same paper [31], the potential is shown to be a rapidly decaying function of $r_{m}^{\prime }$ ($V\left(r\prime_{m}\right)$ change appreciably in scales of the order of $R\times 10^{-1}$), such that $\left|R-r_{m}^{\prime }\right|\ll R∼\left|R-r_{m}^{\prime }\right|\gg R$. Then, expansion for each of those limits (i.e., planar and long-range limit) will have the same functional form, the same inverse cubic potential holds $V\left(z\right)$, which is from (Equation 1):(2)$$V\left(z\right)=-\left(\frac{4\pi }{3}\varepsilon \sigma^{6}\right)\frac{\rho_{s}}{z^{3}},$$
where $z=\left|R-r_{m}^{\prime }\right|$ and $\rho_{s}$ are the density at the interface. In equilibrium (i.e., where average of observables can be locally defined) at the $\tau_{e}$ scale, Boltzmann–Gibbs distribution holds and, assuming that $V\left(z\right)<<k_{B}T$ (i.e., valid for relevant distances scales in Brownian experiments [15]), then the density profile $\rho \left(z\right)$ of the fluid in the gaslike state is given by
(3)$$\frac{\rho \left(z\right)}{\rho_{s}-\rho_{L}}=\exp\left(-\frac{V\left(z\right)}{k_{B}T}\right)≅1-\frac{V\left(z\right)}{k_{B}T},$$
(4)$$\frac{\rho \left(z\right)-\rho_{L}}{\rho_{s}-\rho_{L}}={\left(\frac{\sigma }{z}\right)}^{3},$$

Here, *T* is the temperature of the system, $k_{B}$ the Boltzmann constant, and $\rho_{L}$ is the density in the incompressible liquid phase. Further, as stated above, the appreciable change is in the order of $R\times 10^{-1}$ and *R* dictates the validity of the approximation (Equation 3). As an example, if *R* is in the order of 50 nm, then the approximation performed to obtain Equation (Equation 4) and, therefore, Equation (Equation 4), is expected to hold for distances on the order of 0.1μm up to infinity and the density is expected to change appreciably in the order of $10^{-7}$ m, (i.e., which means it can be assumed constant for smaller scales/subdomains such as $10^{-8}$ m or $R\times 10^{-2}$ m, this is the scale of local equilibrium), which are the relevant experimental scales [15].

We are interested in seeing how Equation (Equation 4) generated by cluster 1 interacts with cluster 2 (in the end, we will see how this changes the displacement probability distribution). For these purposes, the relevant distance that we are interested in the behavior of the fluid is $z^{\prime }=z+R=\langle r^{\prime }\rangle -R+R$. This is where the fluid interacts with the cross-section of a different cluster; then, $z^{\prime }=\langle r^{\prime }\rangle$. This means that *z* can be replaced by $\langle r^{\prime }\rangle$ (i.e., mean relative distance measured from the center of a cluster). The $\rho_{L}$ and $\rho_{s}$ terms are there to satisfy the boundary conditions. When r→∞, then $\rho_{L}$ (i.e., density of the fluid in the gel-like phase) is recovered as one should expect. When $\langle r^{\prime }\rangle =R$, the density of the fluid at the interface (in the gaslike state) is recovered and is called $\rho_{s}$. Note that in this case, $\rho_{s}{<\rho }_{L}$; therefore, density increases up to infinity following Equation (Equation 4).

The relevant density differences in (Equation 4) are $\tilde{\rho }=\rho \left(\langle r^{\prime }\rangle \right)-\rho_{L}$ and $\rho_{b}=\rho_{s}-\rho_{L}$. Equation (Equation 4) is
(5)$$\tilde{\rho }=\rho_{b}{\left(\frac{\sigma }{\langle r^{\prime }\rangle }\right)}^{3},$$

Since there is radial symmetry, Equation (Equation 5) can be further simplified, given that
(6)$$\tilde{\rho }=\frac{dN}{dV}=\frac{dN}{d\langle r^{\prime }\rangle }\frac{1}{\left(4\pi {\langle r^{\prime }\rangle }^{2}\right)},$$
(7)$$\rho_{b}=\frac{1}{\left(4\pi \sigma^{2}\right)}{\frac{dN}{d\langle r^{\prime }\rangle }|}_{\langle r^{\prime }\rangle =R};$$
notice that ${\frac{dN}{d\langle r^{\prime }\rangle }|}_{\langle r^{\prime }\rangle =R}=-\frac{1}{\sigma }$. Therefore,
(8)$$\frac{dN}{d\langle r^{\prime }\rangle }=-\left(\frac{1}{\langle r^{\prime }\rangle }\right),$$

Equation (Equation 8) was obtained by treating the fluid as an ideal gas. However, it is required in this model to consider molecular interactions in the gas (Van der Waals gas). This type of gas can be treated as an ideal gas with a modified number of particles ZN, where *Z* is the compressibility factor [33] (note that *Z* is a ratio of volumes and it is a nondimensional quantity). Z<1 means attractive interactions, Z=1 means no interaction (ideal gas), and Z>1 means repulsive interactions between fluid molecules. Then, Equation (Equation 8) simply becomes
(9)$$\frac{d\;(ZN)}{d\langle r^{\prime }\rangle }=-\left(\frac{Z}{\langle r^{\prime }\rangle }\right),$$

It is this radial gradient of the number of particles that generates a radial attractive interaction between two clusters. As a comment, since the derivation of (Equation 9) was performed in the realm of classical statistical mechanics and local equilibrium was assumed, the quantity $\langle r^{\prime }\rangle$ was defined with angle brackets, since it represents an average and is a local observable in the $\tau_{e}$ scale. Note that the number of particles gradient in (Equation 9) exists only in the radial direction. Further, the difference d(ZN) is taken with respect to the number of particles the fluid would have in absence of the Lennard-Jones potential.

Moreover, the number of particles gradient in (Equation 9) do not generate any force acting on the same cluster that is producing it (self-interaction). Of course, due to its spherical symmetry, it can be deduced that this force is zero. However, it does generate a force acting on a different cluster (pairwise interaction). This force is derived as follows:

For the following, the thermodynamics for a classical ideal gas is used. This is justified since local equilibrium is assumed at the scales we are working with. At any given time, consider that cluster 1 is located a mean radial relative distance $\langle r^{\prime }\rangle$ from cluster 2. (In the following, it will be assumed that cluster 1 is in the reference frame of cluster 2). As it is done in the derivation of mean free path with ideal gases [34,35], the cluster will be treated as a point particle with cross section of $\pi R^{2}$, where *R* is the radius of the sphere. Now, imagine a cylinder enclosing the cluster 1 cross-section oriented in the radial direction with the same cross-section as the cluster and height $d\langle r^{\prime }\rangle$ (see Figure 2). This cylinder will have a greater number of particles ${ZN}_{1}$ in its farthest half of cluster 2 (the fluid is not perturbed by cluster 2 for this half; then, the density remains unchanged) than in its closest half ${ZN}_{2}$ (see Figure 2b). This means that the force acting on cluster 1 will be simply due to the difference in the number of particles and the ideal gas law (Considering *Z*), and will be acting radially inwards towards cluster 2, as it can be seen from Figure 2a below:

Then, the magnitude of this force (i.e., interaction force) that is due to the local difference in the number of particles (analogous to two gases with different numbers of particles hitting a large wall/interface from each side (i.e., cluster 1 cross-section)) is given by
(10)$$F_{n}=\frac{Z\left({|N}_{2}-N_{1}|\right)}{d\langle r^{\prime }\rangle }k_{B}T=\frac{k_{B}T}{\left|\frac{d\langle r^{\prime }\rangle }{d\left(ZN\right)}\right|},$$

Now, using Equation (Equation 9),
(11)$$\left|\frac{d\langle r^{\prime }\rangle }{d\left(ZN\right)}\right|=\frac{\langle r^{\prime }\rangle }{Z},$$

Note that $\langle r^{\prime }\rangle$ is not the root-mean-squared displacement, but rather the root-mean-squared relative distance. Finally, using (Equation 10) and (Equation 11), the magnitude of the force is
(12)$$F_{n}=\frac{k_{B}T}{\frac{\langle r^{\prime }\rangle }{Z}},$$

Note that Equation (Equation 12) is dependent on *Z* and the quantity $\frac{\langle r^{\prime }\rangle }{Z}$ defines the expected range for which this force is relevant. This is to be determined experimentally (a critical displacement $\frac{\langle r^{\prime }\rangle }{Z}$, where *Z* suffers a discontinuity, due to the phase separation assumed in this model) and it has been reported in some situations to be up to the order of a few μm [15,19]. Further, note that the gradient of number of particles in Equation (Equation 9); the density gradient; and therefore, the heterogeneity of this weakly interacting fluid is expected to be relevant up to the same scale of *R*. That is, unlike an ideal incompressible fluid, which would be homogeneous in density at these scales. The force $\vec{F}$ that corresponds to (Equation 12) is given by (considering the negative sign in (Equation 9))
(13)$$\vec{F}=-\frac{k_{B}T}{\frac{\langle r^{\prime }\rangle }{Z}}\hat{r^{\prime }}\;,$$
where $\hat{r^{\prime }}$ is the unit vector in the direction of the relative distance between two clusters $r^{\prime }$. This force can be included in Newton’s second law in vector form and introducing the shorter time scale τ previously mentioned. The Langevin approach is presented as follows:(14)$$m\ddot{\vec{r}}=-\alpha \dot{\vec{r}}-\frac{k_{B}T}{\frac{\langle r^{\prime }\rangle }{Z}}\hat{r^{\prime }}+\varepsilon \left(\tau \right),$$
where all the derivatives are taken with respect to τ. Further, $\vec{r}$ is the displacement vector of the cluster as measured from the origin. The first term to the right of the equals sign represents the viscous drag force generated by the fluid (with α being the drag coefficient), the second term is the density gradient force introduced in this model, and the third term $\varepsilon \left(\tau \right)$ is the white Gaussian noise [36] generally assumed in Brownian Motion. In the overdamped case (low-Reynolds regime), the inertial term (with *m* being the mass of the diffused particle) in the left-hand side of the equation is zero. Then, the simplified Newton’s equation only in the radial relative direction $r^{\prime }$ is
(15)$$0=-\alpha \dot{\vec{r^{\prime }}}-\frac{k_{B}T}{\frac{\langle r^{\prime }\rangle }{Z}}\hat{r^{\prime }}+\varepsilon \left(r^{\prime },\tau \right),$$

Note that since the force in (Equation 13) acts only in the radial direction, in the other two dimensions perpendicular to $r^{\prime }$ the motion will be a regular Brownian motion. Further, in these two other dimensions the particles distribution at local equilibrium is uniform, which means there exists spherical symmetry (i.e., the particle distribution translates to the random motion of a sphere in the radial direction). Additionally, since we are dealing with nonaccelerated frames, ${\vec{r}}^{\prime }$ can be replaced by $\vec{r}$ (i.e., displacement measured from the origin). In terms of the magnitude *r* (i.e., distance from the origin to a cluster), Equation (Equation 16) is
(16)$$0=-\alpha \dot{r}-\frac{k_{B}T}{\frac{\langle r^{\prime }\rangle }{Z}}sgn\left(r\right)+\varepsilon \left(r,\tau \right),$$

Note that the magnitude of the second term to the right is a constant since local equilibrium is assumed and $\frac{\langle r^{\prime }\rangle }{Z}$ is a constant observable (as a comment, this also means that for the proposed stochastic process white Gaussian noise can be used [36], since inside this local equilibrium domain, density is uniform). The process in (Equation 16) has already been studied in [37,38,39,40]. There are a couple of remarks here. First, the propagator (i.e., initial condition given by delta function centered at $r_{0}$) as solved in [38] by setting up the corresponding Fokker–Planck equation to (Equation 16) is given by
(17)$$P\left(r,\tau \right)=\frac{e^{-\frac{\tau }{4}}}{2\sqrt{\pi \tau }}exp(\frac{-\left|r\right|+\left|r_{0}\right|}{2}-\frac{{\left(r-r_{0}\right)}^{2}}{4\tau })+\frac{\exp\left(-\left|r\right|\right)}{4}\left[1+erf\left(\frac{\tau -\left|r\right|-\left|r_{0}\right|}{2\sqrt{\tau }}\right)\right],$$

According to [39], a time-dependent diffusivity can be defined, and a modified form of the fluctuation–dissipation theorem holds, where temperature depends on time τ (i.e., out of equilibrium process). Further, in this time scale, the process is instantaneously diffusive [39]. Then, when τ→t, the fluctuation–dissipation theorem holds, where an average in time τ is taken and the diffusivity and temperature are replaced by averages. This means the diffusivity *D* is given by
(18)$$D=\frac{k_{\beta }T}{6\pi \eta R},$$
where *D* and *T* are average in the time scale τ, and observables in the time scale $\tau_{e}$. Since this process is diffusive, one has that the root-mean-squared radial displacement of a single cluster is $\sqrt{2Dt}$ (there is only one degree of freedom; therefore, it is not dependent on dimension). Moreover, τ has been replaced by *t*. This is the long time limit and, for example, (as an estimation) if $R∼10^{-9}$ m typically $\tau_{e}∼100$τ. (Note that the use of *t* in equations is for practical purposes as *t* denotes the experimentally measured time and $\tau_{e}=t$, as defined above). Given that local equilibrium is assumed in the $\tau_{e}$ scale, the process updates adiabatically. The mean relative displacement to a different cluster would require including a factor of $\sqrt{2}$ as is shown in (Equation 19) [34]. Finally, the root-mean-squared relative distance traveled as measured from cluster 2 is required (therefore, a factor of $\frac{1}{2}$ is also included in (Equation 19).
(19)$$\frac{\langle r^{\prime }\rangle }{Z}=\frac{\sqrt{2}\sqrt{2Dt}}{2Z}=\sqrt{\left(\frac{D}{Z^{2}}\right)t},$$

The stationary solution to the Fokker–Planck equation corresponding to (Equation 16) is the local equilibrium solution and is given by the Boltzmann–Gibbs distribution [38]. The argument of the Boltzmann–Gibbs distribution is the potential $V\left(r\right)$ derived from the force in (Equation 13), which is
(20)$$V\left(r\right)=\frac{k_{B}T}{\sqrt{D_{eff}\;t}}r,$$
where $D_{eff}=\frac{D}{Z^{2}}$. Now, the Boltzmann–Gibbs distribution is
(21)$$p\left(\vec{r}\right)={Ae}^{-\;\frac{V\left(r\right)}{k_{B}T}},$$
where *A* is a constant. Note that the potential is linear in the absolute value of the position. The full stationary probability distribution in the radial direction is, therefore (the $k_{B}T$ term cancels),
(22)$$p\left(\vec{r},t\right)\propto e^{-\frac{r}{\sqrt{D_{eff\;}\;t}}},$$
with $r=\left|\vec{r}\right|$. The expression in (Equation 22) is valid from 0 up to $t_{c}$, where, as mentioned at the beginning of this section, there is some $\langle r^{\prime }\rangle_{c}$ (with $\langle r^{\prime }\rangle_{c}=\sqrt{D_{eff\;}\;t_{c}}$. Note that $D_{eff\;}$ suffers a discontinuity here) or, correspondingly, some $t_{c}$ (should be determined experimentally and could be a very large value) at which the gaslike phase ends abruptly and the liquid phase starts. In general, in these types of phase separations, the compressibility factor *Z* suffers a vast discontinuity [30]. It generally jumps from a value close to 1 to a very small value [30] that is reminiscent of a liquid. Of course, this indicates that interactions between molecules in the liquid phase are a lot stronger than in the gaslike phase [33]. Now, since *Z* has a very large decrease, $D_{eff}$ has a very large increase. It (as specified in the assumptions of this model) can be realized that the magnitude of the force in (Equation 13) will be several orders of magnitudes less (high $D_{eff}$) than the viscous drag force term in (Equation 16). This means, for this regime, that the force in (Equation 16) can be safely ignored. After $t_{c}$, the Langevin equation becomes
(23)$$\alpha \dot{\vec{r}}=\varepsilon \left(t^{\prime }\right),$$
or the corresponding Fokker–Planck equation being the diffusion equation [41]
(24)$$\frac{\partial p_{\left(\vec{r},t^{\prime }\right)}}{\partial t^{\prime }}=D_{effL}\nabla^{2}p\left(\vec{r},t^{\prime }\right),$$
which should be solved with (Equation 22) evaluated at $t_{c}$ as initial condition and $t^{\prime }=t-t_{c}$. To account for the discontinuity of *Z*, the effective diffusivity for this regime will be denoted by
(25)$$D_{effL}=\frac{D}{{Z_{L}}^{2}},$$
and the effective diffusivity in the gaslike phase will be denoted by
(26)$$D_{effG}=\frac{D}{{Z_{G}}^{2}},$$

Since $D_{effG}<<D_{effL}$, Equation (Equation 22) can be seen as a delta function when evaluated at $t_{c}$ and used as an initial condition to solve (Equation 24), meaning that after $t_{c}$ the probability density will have some transition probability that will evolve rapidly towards a Gaussian, which is
(27)$$p\left(\vec{r},t^{\prime }\right)∼\frac{1}{{\left(4\pi D_{effL}t^{\prime }\right)}^{\frac{3}{2}}}e^{-\frac{r^{2}}{4D_{effL}t^{\prime }}},$$

From now on, the probability distribution in (Equation 22) will be denoted as $p_{G}\left(\vec{r},t\right)$ (gaslike phase) and the full solution to (Equation 24) with the initial condition being $p_{G}\left(\vec{r},t_{c}\right)$ will be denoted as $p_{L}\left(\vec{r},t^{\prime }\right)$(liquid phase).

The description above gives an accurate time evolution of the probability distribution of the position of the clusters for the two different regimes (the gaseous regime with $D_{effG}$ and the liquid regime with $D_{effL}$). However, it does not tell you which probability distribution you should use at any given time (over a single scale over a time interval *t*) $p_{G}\left(\vec{r},t\right)$ or $p_{L}\left(\vec{r},t^{\prime }\right)$. This means that, at time $t<t_{c}$, a cluster may be located at a position $\vec{r}$ such that it is either in the gaslike phase or in the liquid phase. The same happens at $t>t_{c}$. This means that the gaslike and liquid phases can be regarded as two possible states of the system, with no preference over one or the other. Of course, this should be accounted for when taking averages of observables, one should take the average over all possible states. Therefore, the mean squared displacement for this process will be given by
(28)$$\langle {r\left(t\right)}^{2}\rangle =0.5\left(\int_{-\infty }^{\infty }{{\left|\vec{r}\right|}^{2}p}_{G}\left(\vec{r},t\right)d\vec{r}+\int_{-\infty }^{\infty }{{\left|\vec{r}\right|}^{2}p}_{L}\left(\vec{r},t\right)d\vec{r}\right),$$
which the simplifying of gives
(29)$$\langle {r\left(t\right)}^{2}\rangle =0.5\left(6D_{effG}t+{6D}_{effL}t\right),$$

Rearranging terms,
(30)$$\langle {r\left(t\right)}^{2}\rangle =6\left[0.5\left(D_{effG}{+D}_{effL}\right)t\right],$$
where $D_{avg}$ can be seen as an average diffusivity and is given by
(31)$$D_{avg}=0.5\left(D_{effG}{+D}_{effL}\right),$$

Equation (Equation 30) can be rewritten as
(32)$$\langle {r\left(t\right)}^{2}\rangle =6D_{avg}t,$$

Equation (Equation 32) leads to interpret the whole Brownian process discussed until now in two different regimes with diffusivities $D_{effG}$ and $D_{effL}$ as a single rescaled process, such that the standard deviation of the PDF is given by (Equation 32).

Finally, from (Equation 32) and the analysis up until before Equation (Equation 28), it can be inferred that the observed probability distribution of position $\vec{r}$ of the cluster before $t_{c}$ will be given by the normalized Equation (Equation 22) and after $t_{c}$ by the normalized solution to (Equation 24) with the initial condition of (Equation 22). Then, both probability distributions should be rescaled such that their mean squared displacement (i.e., standard deviation squared) at any given time is given by (Equation 32).

## 3. Initial Theoretical and Experimental Considerations

A few initial remarks concerning the proposed model and some relevant experimental observations are summarized here. In experiment [15], the Brownian motion of colloidal beads on phospholipid bilayer (DLPC) tubes (1-D) was studied. Lipid bilayers tend to have weak intermolecular interactions between “lipids tails” [42]. (London dispersion forces are the type of molecular interactions occurring between lipids). The weaker these intermolecular interactions, the more flexible the bilayers (i.e., membranes) and vice versa [42]. Therefore, (also motivated by the role of weak interactions in phase separation in different studies such as [29]), it is suggested here that the Brownian particle–fluid long-range interactions can lead to a phase separation and the whole machinery developed above can be applied.

First, note that what is being measured in the experiment of phospholipid bilayer tubes [15] (with no cholesterol, where Brownian yet non-Gaussian diffusion is reported) is $D_{avg}$. Then, according to our model, we can write this quantity as
(33)$$D_{avg}=0.5\left(\frac{D}{{Z_{G}}^{2}}+\frac{D}{{Z_{L}}^{2}}\right),$$

Approximating the lipids tails in the gaslike phase as an ideal gas (i.e., almost no interaction), meaning $Z_{G}∼1$, then
(34)$$D_{avg}=0.5\left(D+D_{effL}\right),$$

As mentioned before, $Z_{L}$ is a small number. This means $D_{avg}\gg D$, which is what was found in the experiment [15] (i.e., unusually high diffusivity).

The Lennard-Jones potential introduced in this model causes a reduction in local pressure in the liquid (meaning the local pressure is less than the external pressure at which the experiment is being conducted). This means that determination of $Z_{L}$ requires full knowledge of the effective local pressure experienced in the liquid at the phase separation point, which is not an easy task to estimate theoretically; therefore, a direct calculation of $D_{avg}$ is not performed here. (However, it could in principle be done). In any case, the value of $D_{avg}=0.40\;\mu$m${}^{2}\times$s${}^{-1}$ and *D* reported in this experiment [15] can be used to calculate the value of $Z_{L}$ and check whether it is a reasonable value for a liquid at room temperature, low pressures, and far from the critical point. *D* is calculated by using Einstein’s relation:(35)$$D=\frac{k_{B}T}{6\pi \eta_{e}R},$$
where *T* is temperature, $k_{B}$ is Boltzmann constant, *R* is the radius of the Brownian particle, and $\eta_{e}$ is the extensional viscosity of the fluid. Extensional viscosity considers shear viscosity and bulk viscosity, which should be the case for a compressible fluid [43,44]. For a Newtonian fluid, $\eta_{e}$ is given by [44]:(36)$$\eta_{e}=3\eta ,$$
where η is the shear viscosity of the fluid. Using (Equation 35) and (Equation 36),
(37)$$D=\frac{k_{B}T}{18\pi \eta R},$$

## 4. Application of the Model

In order to compare model predictions with experimental results, experimental data taken from Wang et al. [15] are employed. In this experiment, the temperature is reported to be T=22
${}^{{}^{\circ}}$C, the viscosity of the media η is reported to be ≈100 times higher that than of bulk water, and the radius of the Brownian particle is R=50 nm. Therefore, with the conditions of this experiment, Equation (Equation 37) is used to find that D∼0.014μm${}^{2}\;\times$ s${}^{-1}$. Further, using this result and Equations (Equation 25) and (Equation 34), it is calculated indirectly that $D_{effL}∼0.79$
μm${}^{2}\;\times$ s${}^{-1}$ and $Z_{L}∼0.13$ (note that $Z_{L}$ is a reasonable value for a liquid at room temperature and low pressures [33]). Further, note that $D_{avg}\gg D$, since 0.40≫0.014.

Regarding the temporal evolution of the probability distribution, in this experiment, $t_{c}∼4s$. This means that up until t=4 s, the probability density function (PDF) in the displacement x will be given by the analogous 1D expression to (Equation 22) (i.e., Laplacian distribution) with a diffusivity of $D_{avg}$. Starting at t=4 s, a numerical implementation of the proposed model in Mathematica shows that after ∼2 s (i.e., t∼6 s), the PDF (Equation 22) had transitioned towards a 1D Gaussian in the displacement x (see Figure 3a,b below). This is precisely what was observed in this experiment [15].

A brief description of the figures and data is given here. The solid line curves of Figure 3a below were obtained from the proposed model. At t=60 ms, t=0.6 s, and t=3 s, the corresponding 1D un-normalized Equation (Equation 22) was used. It was rescaled such that its mean square displacement is given by (Equation 32) 1D analogous (i.e., $\langle {r\left(t\right)}^{2}\rangle =2D_{avg}t$) with $D_{avg}=0.40\;\mu$m${}^{2}\times \;$s${}^{-1}$.

Furthermore, at t=5.8 s, the differential Equation (Equation 24) in 1D with $D_{effL}∼0.79\;\mu$m${}^{2}\;\times$ s${}^{-1}$ was solved numerically in Mathematica. The initial condition was taken to be the 1D un-normalized Equation (Equation 22) evaluated at time $t_{c}=4$ s (which is the critical time reported by Wang et al., where the Laplacian distribution starts changing towards a Gaussian) with diffusivity $D_{effG}∼0.014\;\mu$m${}^{2}\;\times \;$s${}^{-1}$. The solution was rescaled such that $\langle {r\left(t\right)}^{2}\rangle =2D_{avg}t$. As a result, the transition probability $p_{L}\left(x,t\right)$ at t=5.8 s is found to be
(38)$$p_{L}\left(x\right)\propto \;\;\;0.5e^{35.55-0.45x}\left\{erfc\left(5.96-0.038x\right)\;+e^{0.91x}erfc\left(5.96+0.038x\right)\right\},$$
where *x* is the 1-D displacement of the bead and
(39)erfc(w)=1−erf(w),

It is interesting to note that (Equation 38) plotted in Figure 3a agrees very well with the experimental data points. With respect to the experimental data points, they were obtained from [15]. Linear regression analysis was performed to fit the experimental data points in Figure 3a. By fixing the lowest-order parameter, a calculation of the percentage error of the highest-order coefficient/parameter between the theoretical prediction and the experimental best fit was performed at the corresponding times. At t=60 ms, the percentage error was found to be 0.28%; at t=0.6 s, the percentage error was found to be 2.0%; at t=3 s, the percentage error was found to be 2.0%; and at t=5.8 s, the percentage error was found to be 3.4%.

From Figure 3b, it can be inferred that at some point before 200 d, the slope of the brown and gray lines changes. Note that this figure is plotted in a log–log scale, which means the graph shows the log(log(p(x,t))) in the y-axis vs. the $log(\frac{x}{d})$ in the x-axis. This indicates exponential decay for the time passed after $t_{c}=4\;s$ of t=1s and t=1.5s. At t=2s, the slope of the red line is constant, which indicates Gaussian behavior for all *x* up to x=200d.

From the figures above, it can be concluded that the theoretical model agrees with the experiment. Percentage error in the parameters do not exceed 3.5%. Further, besides the theoretical prediction of the emergence of a Laplacian distribution and an eventual Gaussian distribution in the particle displacements, it is remarkable that the transition probability at t=5.8 s and the transition time (which emerge from ideal scenarios) are an accurate representation of the experimental data.

Finally, as a second part of the experiment, the membranes of the fluid were filled with cholesterol. Everything else was held constant and no exponential distribution was observed. As it has been observed before, cholesterol provides rigidity to the membranes by strengthening molecular interactions between lipids [45]. Since not enough energy is provided to break those interactions by the interaction Brownian particle-fluid (via the Lennard-Jones potential proposed in this model), no phase separation occurs and one is in the regime of “regular Brownian motion”. Further, in this case, the diffusivity was observed to be D=0.012
μm${}^{2}\;\times$ s${}^{-1}$. This should be the case, since by using Einstein’s relation (Equation 37) and considering the 20% increase in viscosity due to cholesterol [15], the theoretical value of the diffusivity is exactly 0.012
μm${}^{2}\;\times$ s${}^{-1}$.

## 5. Conclusions

A new model to explain Brownian yet non-Gaussian behavior is proposed, by including molecular interactions. Two regimes in a weakly interacting and highly compressible fluid are presented. Long-range interactions Brownian-particle fluid cause a phase separation for short distances. In the first regime, the fluid behaves similar to a real gas interacting with an interface with no shear viscosity. If only pairwise cluster interaction via the fluid is considered, a Laplace distribution on the displacements of a phase-separated cluster is derived. In the second regime, the fluid behaves as a liquid, but with very high diffusivity (i.e., due to low compressibility/high mean free path caused by the separation of fluid particles via Lennard-Jones potential). The Laplace distribution evolves starting at a time $t_{c}$ via a transition probability (Equation 38) rapidly into a Gaussian. (All the pdfs are rescaled, such that $\langle {r\left(t\right)}^{2}\rangle$ is given by the 1D analogous of (Equation 32), since the cluster might be in either the gaslike or in the liquid phase at some point in time).

The experimental data obtained by Wang et al. are used to apply the model. It is shown that the theoretical model explains via a phase separation the physical cause for an unexpectedly high diffusivity. In addition, the lack of a phase separation when using cholesterol and the compressibility and shear deformation of the liquid (i.e., considering extensional viscosity) leads to a different and smaller value of the diffusivity that fully agrees with the experimentally reported value. Further, the pdfs for different times predicted by the model are compared with the experimental data and are found to be in excellent agreement. At $t_{c}=4$ s, the pdf starts transitioning away from being exponential and, after approximately 2s, is found to be Gaussian, as reported in the experiment.

Finally, this approach links theoretical concepts in the previous research of fluid behavior [44,46,47] and theory of interfaces [31] to Brownian motion, which indicate that there are strong motives to expand studies of highly compressible liquids (in colloidal suspensions) and their molecular interactions with a Brownian particle (i.e., different effective/interaction potentials to the one in (Equation 2) can be proposed to model different regimes at shorter time scales, where different types of diffusion processes may occur). Then, these effects can be incorporated into Langevin dynamics without the need of complicated mathematical modifications to noise terms.

## Figures and Tables

**Figure 1 materials-15-05808-f001:**
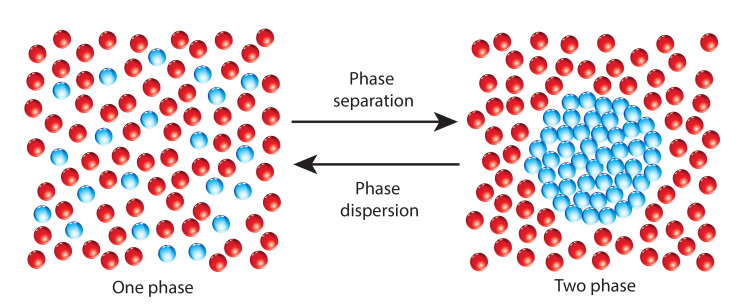
To the **left**, a biological fluid in a dispersed state is depicted. The fluid behaves as an ideal liquid. To the **right**, the same fluid in a liquid–liquid phase separation is depicted. The phase in blue is highly compressible. The other phase in red is highly incompressible.

**Figure 2 materials-15-05808-f002:**
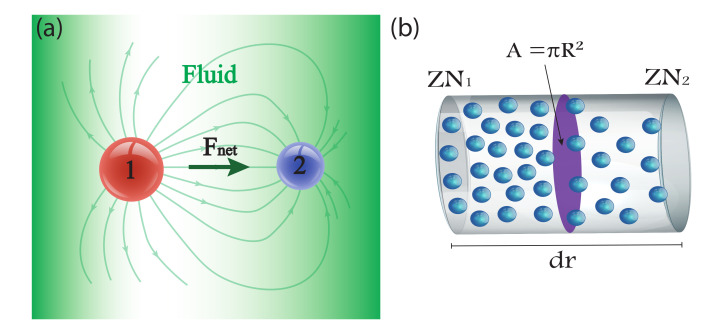
(**a**) Pairwise interaction of 2 clusters via the radial force field in Equation (Equation 13). (**b**) Volume element that shows the gradient in the number of fluid particles (fluid particles in blue and cross-section of cluster 1 in purple) that generates a force acting on cluster 1.

**Figure 3 materials-15-05808-f003:**
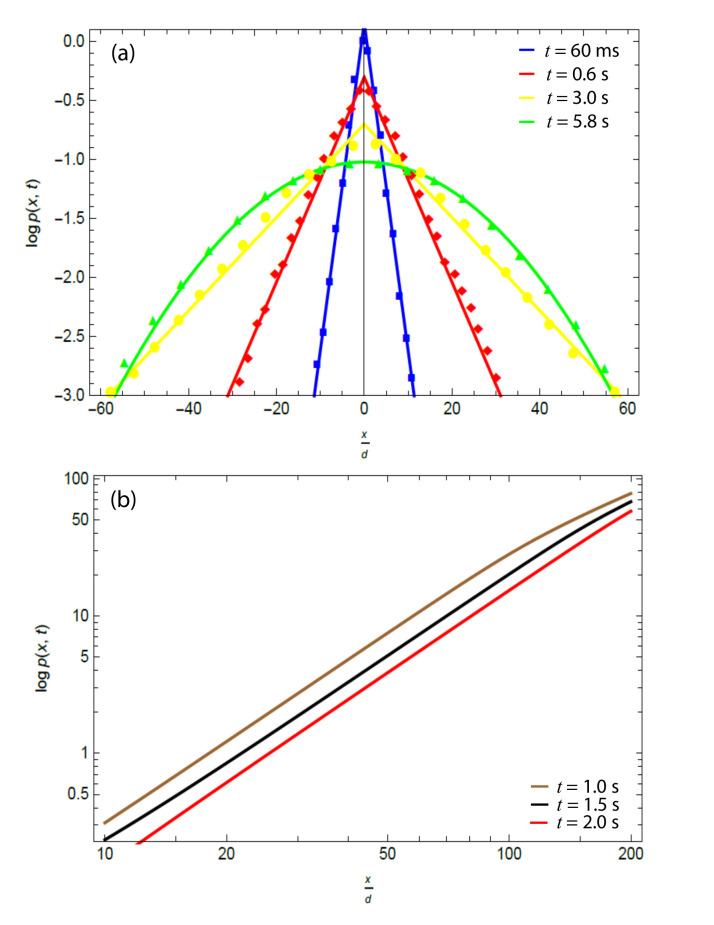
(**a**) Solid line curves obtained from the model and experimental data points of the logarithm of the displacement probability distribution plotted against particle displacement normalized by the particle diameter *d* at different times. (**b**) Theoretical prediction of the logarithm of the displacement probability distribution plotted against particle displacement normalized by the particle diameter *d* at a log–log scale at different intervals of times. These intervals of times are measured with respect to an initial time of $t_{c}=4$ s.

## Data Availability

Data and the code are available from the corresponding author on request.

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
