# Peer review of "A Novel Physical Mechanism to Model Brownian Yet Non-Gaussian Diffusion: Theory and Application"

_materials, 2022, doi:10.3390/ma15175808_

Round 1

Reviewer 1 Report

The manuscript ‘Understanding Brownian yet non-Gaussian diffusion via

long-range molecular interactions’, presents the theoretical modelling of the distance between two phase separated group of molecules called ‘Brownian particle’ experiencing long-range interaction. Main result is that the distribution of the distance between the two ‘Brownian particles’ shows a Laplace distribution at short time and converges to a Gaussian distribution at longer time similar to the very popular ‘Brownian yet non-Gaussian’ (BnG)  motion. The subject of the manuscript is interesting as it presents a previously undescribed physical mechanism responsible for BnG motion. However, the manuscript suffers from its lack of clarity that makes some physical arguments difficult to understand (at best).  It is a shame to bring innovative ideas in a research field and explain it poorly. I would recommend the manuscript for publication after serious improvement of the text.

I noted many points, but certainly not all, here are some of them:

Here ‘Brownian particle’ is used to refer to a phase-separated group of molecules. This is very confusing, a particle should be a single unit, I strongly disapprove its use here, I would suggest ‘cluster’ or ‘aggregate’. That would also avoid sentences about the number of particles in the Brownian particle.

Line 34: ‘The Brownian yet non-Gaussian diffusion reported byWang et al. is based in the classical random walk in which mean-square displacement is simply proportional to time but, instead, has an exponential behavior with the decay length of the exponential being proportional to the square root of time.’

There is a problem in this sentence. How does MSD have exponential behavior?

Line 82: ‘gaseous state fluid’: I would be very surprised to see a biological membrane behaving as a gas.

Line 75: a phase separation occurs [26], otherwise no phase separation occurs,

Line 86: ‘the t scale’

It is weird to use the universal variable for time to here be a timescale.

Line 96: ‘This model was already studied for a spherical shell interface in and it was found that for the planar limit [29] at’

Reference is misplaced.

Line 104: ‘|R-r|<<R  |Rr|>>R’

Do I need to chose which condition I prefer?

Line 116: ‘which means it can be assumed constant for smaller scales/sub-domains such as 108, or R×102’

What are the units?

Line 118 ‘As a comment a similar derivation of equation (4) was done by De Gennes in vapour liquid interfaces [29].’

Going rapidly over De Gennes’ article, from Eq. (1) to Eq.(4) the derivation is exactly the same. I think it should be emphasized at the beginning rather that in the end. Maybe something like: For clarity purpose, we reproduce here the derivation from De Gennes…

Line 121: ‘The interaction of two Brownian particles is of interest.’

Why is it of interest? Could you elaborate more?

Line 135: what are the units of Z?

Line153: ‘it will be assumed that one is in the reference frame of Brownian particle 2’

‘one’ should be replaced with Brownian particle 1.

Line 172: ‘This is to be determined experimentally and it has been reported in some situations to be up to the order of few seconds or a few μm [15,19].’

What is to be determined? Is it Z ? How can this be either seconds or um?

Eq.(17): equation is not written properly

Line210: ‘R10^-9mtypicallyt R 10 100t’

What does this mean?

Line 224: \rangle is missing

Line 276: ‘This means that the Brownian particle-fluid long-range interactions should be 276

enough to cause a phase separation’

Is there any experimental evidence for that?

Line 295: missing reference

Line 309: ‘Numerical implementation of the proposed model in Mathematica shows that after

2s the PDF (22) transitions towards a 1D Gaussian in the displacement x (see figures below).’

I don’t see a change at 2s,  this is contradictory with the previous sentence that says it takes 4s to reach a Gaussian regime.

Figure 2b: I don’t understand this figure, if I take the logarithm of a pdf and it diverges with distance, this is not a good sign..

Line 298: ‘Validation of the model’

I disagree that validating is a correct word here, there are already many models that can reproduce Laplace then Gaussian distribution at long time. Fitting the model does not bring a validation of the present model over the others, to do this one would need to prove that it is the interaction between ‘Brownian particles’ that is responsible for the observed effect.

I would suggest ‘application’

Line 317: ‘At t = 5.8s, the differential equation (24) in 1D with De f f L 0.79um2 ・ s1 was solved numerically in Mathematica. The initial condition was taken to be the 1D unnormalized equation (22) evaluated at time t = 4s with diffusivity De f fG 0.014 um2 ・ s1’

How did you chose these times to change model ? This is very unclear and should be explained much better.

Eq.(38): it is interesting to note that this formula is very similar

Line 356: use ‘Laplace distribution’ instead of ‘double exponential’

Line 360: double exponential transitions via a transition probability rapidly into a Gaussian

Please explain better.

Basic rule: figure should always come after its mention in the text

Please have a read at all the text and improve and clarify.

Reviewer 2 Report

In this manuscript, the authors try to understand the transition from Gaussian to Non-Gaussian distribution of the displacement. Based on the model, which takes into account of the molecular interactions, non-Gaussian normal diffusion is observed. Moreover, the new model can match the experimental data from Wang’s group very well. It is a nice try to interpretate the non-Gaussian normal diffusion behavior. However, the following questions should be addressed.

1)    all the symbols and parameters should be defined.

2)    The title of the subsection 2 and 3 are not good. The title should include what you have written in the whole subsection, also according to the whole manuscript, the title only mentioned about the model, where is the results and discussion?

3)    Eq.14, in the gradient density case, is it possible to employ the Gaussian white noise?

4)    The result the inertia term can be ignored is overdamped case, in the low-Reynolds regime.  

5)    Page 10, the source of the experimental parameters are not so clear, such as D_{effL}, Z_L et.al.

6)    There are some groups have studied the non-Gaussian normal diffusion from the microscopic point of view. Such as Front. Phys. 16(3), 33203 (2021), Prev. Rev. E 97,042122(2018)

So, I think this manuscript is not ready to be published in our Journal.

Reviewer 3 Report

In this manuscript, the authors study the Brownian yet non Gaussian diffusion reported recently in several experiments, presenting a normal diffusion with linear time dependence on the mean square displacement but with a non Gaussian displacement distribution. The authors derive a mathematical model based on De Gennes and Langevin dynamics to understand this Brownian yet non Gaussian diffusion by considering long range molecular interactions between molecules of the Brownian particles and the biological fluid. The phase separation of the biological fluid is the important key to understand the experimental results (Ref. [15]), via the variation of the compressibility factor $Z$. At short time scales, the weakly interacting fluid (gaseous like, with $Z=1$) leads to Laplacian distribution in radial particles displacement with a decay length proportional to the square root of the time. After the phase separation, the liquid becomes nearly incompressible (gel like) with $Z<<1$, explaining the high diffusivity and the transition from Laplacian to Gaussian distribution. The experimental data of Ref. [15] validates this mathematical model both qualitatively and quantitatively.

I read this manuscript with great interest and the model derived by the author is, in my opinion, relevant to a better understanding of Brownian yet non Gaussian diffusion. The mathematical model is clearly explained, and the validation with experimental data is appropriate and make this manuscript accessible also to non-specialists as well. In principle, I do therefore support publication of this work in Materials, however, I have several points listed below which the authors should clarify/comment:

1. I have difficulties to understand the difference between the radii $\sigma$ and $R$ defined in the text as the size of the particle (Van der Waals radius) and the radius of the spherical shell, respectively. However, Eq. (8) is valid if and only if $\sigma=R$. First I understood that $\sigma$ was the size of the molecules of the Brownian particle but it is clearly not. Can the author clarify this point?

2. In the Eqs. (3) and (4), $\rho_L$ is not defined whereas $\rho_s$ is correctly defined as the density of the sphere. From Eq. (4), $\rho_L$ is the density far from the Brownian particle when the limit $z>>\sigma$ is taken, or does it have another meaning?

3. From Eq. (14) to Eq. (16), the authors replace the distance to the center of the Brownian particle (denoted r) by the relative distance between two Brownian particles (denoted r'), and vice versa. The Eq. (14) is correct, written for the acceleration of the Brownian particle (proportional to $\ddot r$), and with the viscous drag force applied on the Brownian particle (proportional to $\dot r$), and the force between the two Brownian particles (implying r'). However, I do not understand why the authors write Eq. (15) in terms of r' (without justification) and write Eq. (16) in terms of r (with a correct explanation). Can the author clarify this replacement of r by r' in Eq. (15)?

4. Eq. (17) is slightly different from the Eq. (2.10) of Ref. [34] with a misprint in the first exponential [exp(-\tau/4)] and with the function erfc(x)=1-erf(x) instead of erf(x) in the last term. Can the author explain this difference?

5. The transition probability at time $t=5.8$, given by Eq. (38), is derived numerically in Mathematica with respect to experimental data points. Can this equation be related to Eq. (17) by fitting the parameters $r_0$ and $\tau$?

During the lecture, I have also seen some typos:

1. l.104 Which inequality should we read: $|R-r|<<R$ or $|R-r|>>R$? Since it is a long range interaction, I suppose the second one is correct.

2. l.177 "considering the negative sign in (11)". Since Eq. (11) does not have "negative sign", do the authors mean Eq. (9)?

3. l.210 We should read "$R \sim 10^{-9}m$" and "$t \sim 100 \tau$".

4. l.295 A reference is missing.

5. In Sec. IV, many $\mu$ symbols for µm are misprinted.

Round 2

Reviewer 2 Report

The authors have answered all my questions proposed in last report.